# In Vitro Induction of T Helper 17 Cells by Synergistic Activation of Human Monocyte-Derived Langerhans Cell-Like Cells with Bacterial Agonists

**DOI:** 10.3390/ijms20061367

**Published:** 2019-03-19

**Authors:** Robert Gramlich, Ehsan Aliahmadi, Matthias Peiser

**Affiliations:** 1Institute of Molecular Biology and Bioinformatics, Charité-Universitätsmedizin Berlin, Corporate Member of Freie Universität Berlin, Humboldt-Universität zu Berlin, and Berlin Institute of Health, Charitéplatz 1, 10117 Berlin, Germany; r.gramlich@uke.de (R.G.); Aliahmadi@gmx.net (E.A.); 2Center for Anesthesiology and Intensive Care Medicine, Department of intensive Care, Medical Center Hamburg-Eppendorf (UKE), Martinistrasse 52, 20246 Hamburg, Germany; 3Department of Surgery and Vascular Surgery, Franziskus Krankenhaus Berlin, University Hospital of Charité Universitätsmedizin Berlin, Budapester Straße 15-19, 10787 Berlin, Germany; 4Department of Pesticides Safety, German Federal Institute for Risk Assessment (BfR), Max-Dohrn-Strasse 8-10, 10589 Berlin, Germany

**Keywords:** Toll-like receptor, skin immune system, dendritic cells, T helper cell, Th17, IL-17, IL-22, IL-9

## Abstract

In the case of epidermal barrier disruption, pathogens encounter skin-resident Langerhans cells (LCs) and are recognized by pathogen recognition receptors such as Toll-like receptors (TLRs). As the majority of microorganisms exhibit more than one TLR ligand, the mechanisms of subsequent T cell differentiation are complex and far from clear. In this study, we investigated combinatory effects on Th cell polarization by bacterial cell wall compounds peptidoglycan (PGN) and lipopolysaccharide (LPS) and by bacterial nucleic acid (DNA). Expression of maturation markers CD40, CD80, HLA-DR and CCR7 and the release of IL-1β, IL-6 and IL-23 was strongly enhanced by simultaneous exposure to PGN, LPS and DNA in LCs. As all these factors were potential Th17 driving cytokines, we investigated the potency of combinatory TLR stimuli to induce Th17 cells via LC activation. High amounts of IL-17A and IL-22, key cytokines of Th17 cells, were detected. By intracellular costaining of IL-17^+^T cells, IL-22^−^ (Th17) and IL-22^+^ (immature Th17) cells were identified. Interestingly, one population of LPS stimulated cells skewed into IL-9^+^Th cells, and LPS synergized with PGN while inducing high IL-22. In conclusion, our data indicates that when mediated by a fine-tuned signal integration via LCs, bacterial TLR agonists synergize and induce Th17 differentiation.

## 1. Introduction

In the initial concept of T helper(Th) cell response, in 1986, a bias of Th1 and Th2 was proposed [1]. Reflecting the complex functions and interactions of adaptive immune cells, meanwhile the existence of further Th cell subpopulations, Th9, Th17 and Th22 was recognized, each equipped with exclusive transcription factors, controlling the expression of surface molecules like chemokine receptors and soluble factors such as cytokines [2]. IL-9 releasing Th9 cells are reported to be involved in inflammatory skin and intestine diseases and exhibit potent antitumor activity [3]. IL-17A, IL-17F, IL-21 and IL-22 are cytokines characterizing Th17 cells. Th17 are crucial cells in autoimmune tissue inflammation observed in rheumatoid arthritis, psoriasis, inflammatory bowel disease, type 1 diabetes and multiple sclerosis [4]. The existence and discrete function of a specific Th22 subpopulation is still a matter of discussion, as their phenotype is IL-22^+^IL-17A^−^ and differs from Th17 cells (IL-22^+^IL-17A^+^ or IL-22^−^IL-17A^+^). Recent studies demonstrate that Th22 cells could develop independently of the Th17 lineage [5]. There is evidence from experimental mouse models and human studies that an increasing number of inflammatory diseases are associated with a distinct pattern of Th17 cell-specific cytokines such as IL-17, IL-23 and IL-22 [4,6,7]. In experimental autoimmune encephalomyelitis and collagen-induced arthritis, Th17 were identified as pathogenic effectors [8,9,10]. Th9 cells, however, were candidates for an important role in allergic airway inflammation [11]. In case of inflammatory skin disorders, a coexpression-model for IFN-γ, IL-4, IL-17, and IL-22 with specific pattern for psoriasis, atopic eczema and allergic contact dermatitis was proposed [12]. Originally, definitions of unique subsets were based on exclusive expression of distinct molecules, but recent data from various studies revealed the reality to be more complex. Instead of fixed populations, the plasticity of T cell differentiation and existence of flexible lineages are currently under discussion [2,13].

Besides being an effector role of Th cells in autoimmune diseases, the adaptive immune system responds to microbial exposure by the emergence of Th subsets. Toll-like receptors (TLR) facilitate the innate immune system to sense conserved molecules (pathogen-associated molecular pattern, PAMP) derived from potentially dangerous pathogens [14,15,16,17]. Dendritic cells (DCs) and Langerhans cells (LCs) exhibit the most complete arsenal of TLRs [18,19,20,21]. Upon receiving a trigger by microbial-derived ligands such as LPS or peptidoglycan (PGN) or stimulation with toxins or irritants (danger-associated molecular pattern, DAMP), DCs and LCs release cytokines, some of which are mandatory to polarize Th-subsets [22,23,24,25]. Even in T cell polarization and priming, the precise function of epidermal LCs is still far from clear. It was preciously shown that specific strains of LPS and PGN could activate monocyte derived Langerhans cell-like cells and MUTZ-LCs that express moderate and high amounts of TLR4 and TLR2, respectively [21,26].

As epidermal LCs die immediately after isolation from skin explants, MoLCs or MUTZ-LCs were commonly used for LC studies in vitro. However, in human in vitro experiments, LCs were found to be crucial for the induction and control of T cell responses. Dependent on the in vitro model used, LCs isolated from skin or LC-like cells generated from monocytes or CD34^+^precursors, these cells were reported to polarize different T cell subsets like Th1, Th2, and Th17. In detail, stimulated epidermal cells and MoLCs exhibited the capacity to induce IL-17A, IL-4, and IFN-γ from memory, but not in naive CD4^+^T cells [27,28,29,30]. In memory CD4^+^T cells from donors suffering from contact allergy, nickel-specific IFN-γ and IL-17 expression was found in proliferating cells restimulated by MoDCs. More pronounced in LCs than in dermal DCs, IL-22^+^Th cells could be polarized from allogeneic, peripheral CD4^+^T cells and naive T cells [31,32]. In various studies on Th polarization, DCs were stimulated by single TLR agonists such as PGN or LPS and were shown to efficiently trigger the differentiation of Th1, Th2, Th17 and Th9 cells [27,33,34,35,36,37]. Consequently, we asked if activation of dual or triple TLR on LCs synergizes or antagonizes on differentiation of Th cells. Here, we demonstrate a specific synergism of PGN, LPS and DNA in elevating HLA-DR, CCR7, IL-1β and IL-6. If cocultured with LPS together with DNA-stimulated LCs, CD4^+^T cells released large amounts of IL-17, whereas in coculture with LPS together with PGN-stimulated LCs, CD4^+^T cells released large amounts of IL-22. No synergistic effect was found for the induction of IL-9. Thus, our data shows graded effects of synergism with bacterial TLR ligands in inducing Th1, Th17 and Th9 cells.

## 2. Results

### 2.1. CD40 Ligand and TLR Ligands Synergize in Elevation of CD86 and CD83

DC and LC-like cells were generated from human monocytes by cell culture in the presence of GM-CSF, IL-4 and TGF-β, stimulated and maturation-associated molecules were analyzed by flow cytometry. In first series of experiments, a TLR agonist was used together with a well known costimulus from T cells, CD40 ligand. The TLR2 agonist peptidoglycan (PGN) and PGN in combination with CD40L enhanced CD86, HLA-DR and induced CD83 (Figure 1). Comparing cells derived from same donor, CD86 and HLA-DR (MFI values) were exclusively enhanced by additional CD40L in MoLCs but not in MoDCs. In both cells, CD83 was further upregulated by additional CD40L comparing to single PGN situation. After stimulations, only the level of CD80 expression increased. Thus, there was a stimulus-dependent synergism in upregulation of CD86 and CD83, not obvious for expression of HLA-DR and CD80. However, there were apparent differences in MoDCs and MoLCs, in that the latter exhibited a rather immature phenotype.

### 2.2. Simultaneous Stimulations of Different TLRs Maximizes Level of LC Surface Molecules and Cytokines

Considering the natural environment of epidermal resident LCs, we further investigated the effects of three TLR agonists as found in vivo after barrier disrupture by direct exposure of the pathogens to immune cells. Therefore, we focused on LCs in the epidermis for practical reasons represented by MoLCs. Compared to single stimulus, co-administration of LPS and PGN strongly increased CD80 and HLA-DR (Figure 2a), CD40 and CCR7 (Figure 2b). Compared to unstimulated cells, DNA decreased HLA-DR. DNA in cooperation with LPS or PGN and LPS together with PGN slightly increased CD80 expression. Furthermore, DNA together with PGN upregulated CD40 and CCR7 compared to single PGN or DNA, DNA with PGN and LPS induced a further shift only in CCR7 compared to dual stimulation. DCs and LCs are capable of secreting IL-1β, IL-6, IL-12 and IL-23, key cytokines in humans driving and maintaining a T helper cell phenotype. To elucidate the potential to release cytokines specific for T cell differentiation we investigated TLR-stimulated MoLCs, isolated from peripheral blood. In our model, human MoLCs express intermediate levels of TLR2, TLR4 and TLR9 (Figure 2c). In accordance with TLR expression data, after stimulation with PGN and LPS, MoLCs secreted IL-1 β, IL-6 and IL-23 (Figure 3). IL-1β, IL-6 and IL-23 were all significantly enhanced if MoLCs were stimulated with PGN and LPS. DNA did not amplify any cytokine measured, but exhibited a modulatory effect if co-administrated.

### 2.3. LPS-Induced IL-1β and IL-12p70 is Increased by polyU/polyI:C

In addition to bacterial DNA, we investigated analogs of ss- and dsRNA, pU and pIC, for the capacity to activate MoLCs. Neither ssPolyU/LyoVec (pU, TLR8)- nor poly I:C (pIC, TLR3) alone stimulated MoLCs to secrete IL-1β, IL-6 and bioactive IL-12p70 (Figure 4). Significant amounts of IL-1β, IL-6 and IL-12p70 were found after stimulation with LPS. Surprisingly, for these Th17- and Th1-driving cytokines a strong increase was observed if pIC was added to LPS. However, PU exhibited a synergism with LPS in inducing IL-12p70.

### 2.4. LPS and DNA Elevate IL-17 in CD4^+^T Cells

Assuming that a combinatory stimulation with distinct TLR agonists very effectively could induce cytokines in MoLCs, we asked if a capacity by the same TLR-stimulated MoLCs could be deduced even for T cell polarization. For an evaluation of conditions more close to in vivo reality, analyses from MoLC:T cell cocultures were performed in the absence of external IL-2, blocking antibodies or costimulatory CD3/CD28 beads. To compare intra- and extracellular expression of IL-17 and IFN-γ, supernatants and cells of the same individual stimulation situation were each analyzed by ELISA and flow cytometry. Even in the absence of agonists, IL-17 and IFN-γ were found in the cocultures of MoLCs with total CD4^+^ cells (Figure 5). More pronounced than PGN, LPS increased secretion of both cytokines. In addition, only LPS enhanced the amount of double positive cells (IL-17^+^IFN-γ^+^). However, this phenomena was related to a decrease of the single positives in the population of IL-17^+^, but not in the population of IFN-γ^+^ cells. Notably, a significant increase of IL-17 secretion was observed in CD4^+^T cells when LPS and DNA were used for stimulation. Comparing flow cytometric to ELISA data, strong intracellular expression of IFN-γ after LPS together with DNA was not accompanied by higher release of IFN-γ in the media. In contrast to the induction of Th1 and Th17 phenotypes, the Th2 specific cytokine IL-4 was found to be decreased after single and combinatory PGN and LPS administration (data not shown).

### 2.5. LPS Induces IL-9, PGN Induces IL-22 in CD4^+^ Cells

Analysis by intracellular staining showed that there were few amounts of IL-22 in the absence of TLR agonists (Figure 6a). PGN induced IL-22 and coacted with LPS in elevating the number of producing cells. A population of T cells with exclusive expression of IL-9 was found after the addition of LPS, and release of IL-9 failed to increase after additional PGN or DNA. However, no IL-4 could be detected in the supernatants of the cocultures that were positive for IL-9 and IL-22 (not shown). These two agonists also induced further Th-cell specific cytokines IL-9 and IL-22 (Figure 6b) in the supernatants of LC:T cell cocultures. High amounts for IL-22 were detected after co-administration of PGN together with LPS, even higher after additional exposure with DNA, IL-9 was induced exclusively by PGN. 

## 3. Discussion

In this study, we demonstrate a synergism of TLR agonists in elevating maturation associated molecules on LCs and T cell polarizing cytokines, released by activated MoLCs. More important, a synergism via TLR stimulation was even found on the level Th cell-specific cytokines, IL-17 and IL-22, thereby linking pathogen recognition via innate to finally adaptive immunity. Initially, we and other groups provided data for different sets of TLR expression in epidermal, CD34^+^- and monocyted-derived LCs [18,21,29,38,39]. All cells with a more or less pronounced LC phenotype exhibit both expression of TLR2 and TLR4, whereas TLR9 expression is found only at an intermediate to low level in LCs and LC-like cells. Here we used a combination of three different microbial ligands to imitate the “true conditions” found in vivo in the case of pathogenic invasion after a epidermal barrier disrupture when a multi-PAMP exposure has to be assumed for local LCs. In this study, LPS, PGN and bacterial DNA were selected for stimulation of their receptors TLR4, TLR2 and TLR9 to address receptors localized within the endolysosomal compartment and at the cell surface as well [17,37]. Engagement of TLR2 together with CD40 on MoLCs preferentially affected the expression of costimulatoy CD86 and maturation marker CD83. Even with high amounts of PGN (20 µg/mL), CD86 and CD83 on MoLCs failed to reach the level of expression that was found on autologues MoDCs. Similar expression level for CD86 and CD83 in MoDCs and MoLCs was, however, observed if CD40L was used as costimulus. CD40L was previously described as a potent positive feedback signal on differentiating T cells acting in cooperation with microbial signals for the induction of IL-12p70 releasing DCs [40,41].

To investigate complementary, synergistic or antagonistic effects on MoLCs, cells were stimulated by LPS, PGN and bacterial DNA. Full maturation in CCR7 and HLA-DR expression was found when all three stimuli were used in combination. Similar to data in this study using CD40L, MoDCs are already reported to reach full maturation (concerning CD80, CD86, HLA-DR) by single poly(I:C), LPS and R848, but need all three agonists to release substantial amounts of IL-12p70 [42]. In that study, no significant secretion of IL-1β was found in MoDC cultures in single stimulations, but synergistic effects were found when using LPS together with R848 or poly(I:C)+R848.

Concordant data were found for IL-1β at co-administration of PGN together with LPS and for IL-12p70, that in our experiments was induced by poly(I:C) with LPS and peaked with additional polyU. Hence, a profound synergism was measured in MoLCs in spite of low to intermediate expression of TLR3 and TLR8 [38,39], receptors of poly(I:C) and polyU. It could be speculated if stimulation with LPS was the prerequisite for sensing a second signal in a different compartment or if a “temporal window” was opened to integrate multiple TLR signals as proposed [42]. In contrast to IL-6 and IL-1β, significant amounts of IL-23 were exclusively detected in response to PGN together with LPS. However, on the level of DC maturation molecules and release of Th1/Th17- polarizing cytokines, a strong synergism of TLR cooperation was observed. Thus, at least valid for MoLCs, possibly even for additional DCs subsets, a stimulation hierarchy and transmission of integrated maturation signals from MoLCs to T cells could be postulated. In consequence, further transfer of graded signals, integrated by DCs, to the level of cytokines, released by polarized Th cells, was investigated. When analyzing single stimuli, LPS and PGN increased secretion of IL-17.

An additive effect was evident for LPS with DNA in supernatant IL-17 and was not further enhanced by additional PGN. Reflecting the experimental condition using no antibodies that inhibit Th1/Th2 generation, amounts of 1.5 ng/mL IL-17 could be assumed to have physiological relevance. In contrast to IL-17, IFN-γ was not affected by combinatory stimulation with TLR ligands. In single situations and combination with LPS, the amount of IL-17^+^IFN-γ^+^ cells was increased, representing a mixed Th1/Th17 phenotype. In the mouse system, IL-17 and RORγt were shown to be increased by TLR2 stimulation with synthetic agonist even in absence of DCs/LCs [43]. However, polarizing cytokines TGFβ/IL-6 together with anti-IFN-γ and anti-IL-4 may substitute antigen-presenting cells under this condition and promote a differentiation level which enables Th17 to be further triggered by TLR ligands.

In the absence of external factors such as cytokines or anti-CD3/CD28, we could generate Th17 by using TLR2 stimulation [28]. In human skin disorders like allergic contact dermatitis, psoriasis and atopic dermatitis varying amounts of Th17 and IL-22 producing cells were found, indicating disease-specific participation [12,44,45]. Further support for the presence of a mixtured Th phenotype in pathological situations is provided by a study on patients with chronic asthma, where CD4^+^T-cell clones with a Th17/Th2 phenotype that also produced IL-22 were identified [46]. Derived from cells of healthy donors, LCs showed a higher capacity to induce IL-22 producing cells from naive T cells than dermal DCs [31]. Assuming that IL-22 is de novo induced in naive T cells by allogeneic MoLCs, we hypothesized that MoLCs might enhance IL-22 release in total CD4^+^ cells if activated by TLR agonists. Concerning IL-22, LPS with PGN and DNA functioned in strong synergism, boosting IL-22 up to 30 ng/mL (Figure 5A). In costaining with IL-9, we found a slight increase in the generation of IL-22, but whether the true high release of IL-22 is related to induction of larger population of IL-22 producing Th cells or elevating IL-22 protein in a constant subpopulation remains to be determined. Of note, LPS induced an increase of IL-9 in T cells as detected by FACS. However, if measured in the supernatant, there was no increase in the release of IL-9. So we postulate that IL-9 is induced by LPS but accumulates within the cell at the protein level and is only released after a further trigger.

Finally, data presented by this study might be relevant for developing new therapies and as adjuvants for potent vaccines using TLR ligands for induction or improvement of T effector cell response [47]. Recently, research on drug discovery several new compounds activating TLR has been undergoing preclinical and clinical evaluation to prevent and cure cancer and provide autoimmunity [48]. Hence, while being strictly dependent on the pathogen species, the status of the patient`s immune system, the option to initiate, fine tune and regulate Th1 and Th17 by selected TLR ligands will be a central aspect in further therapies such as multiple sclerosis and psoriasis.

## 4. Materials and Methods

### 4.1. Ethical Approval

For whole blood samples, we obtained approval by the ethics committee of the Charité-Universitätsmedizin Berlin. Anonymized blood samples were obtained from the German Red Cross blood donation service Berlin with informed written consent from all participants. All studies were in accordance to the Helsinki guidelines. No part of these studies was conducted outside of Germany.

### 4.2. Isolation of Monocytes and Generation of MoLCs

Peripheral human blood of normal donors was obtained by buffy coats from the German Red Cross blood donation service, Berlin. All studies performed were in adherence to the Helsinki guidelines. By depletion of contaminating cells monocytes were magnetically isolated from PBMC (monocyte isolation kit II, Miltenyi Biotec, Bergisch Gladbach, Germany). In some studies, MoDCs and MoLCs were generated from the same donor blood. Cells were cultured in RPMI 1640 supplemented with 2 mM l-glutamine, 100 IU/mL penicillin, 100 µg/mL streptomycin and 10% (*v*/*v*) heat-inactivated FCS (Biochrom, Berlin, Germany) for 6 days with human recombinant GM-CSF (100 ng/mL), IL-4 (10 ng/mL), and TGF-β1 (for MoLCs only, 10 ng/mL, R&D Systems, Wiesbaden-Nordenstadt, Germany) [49].

### 4.3. TLR Stimulation

For cytokine stimulation experiments MoLCs were cultured at 106 cells/mL for 48 h in 24-well-tissue culture plates (Greiner, Frickenhausen, Germany). Cells were stimulated by 20 µg/mL peptidoglycan (PGN) from *Staphylococcus aureus*, 1 µg/mL ultrapure LPS from *Salmonella minnesota*, 5 µg/mL endotoxin-free DNA from *Escherichia coli* K12, 5 µg/mL polyI:C and 5 µg/mL single stranded polyU complexed with LyoVec (InvivoGen, San Diego, USA), 1 µg/mL recombinant CD40L trimer (Bender, Vienna, Austria). After stimulation, MoLCs were extensively washed before coculturing with allogeneic CD4^+^T cells.

### 4.4. Isolation and Culture of Total and Naive CD4^+^ T Cells

Through depletion of contaminating cells, untouched CD4^+^T cells (mean of 97% CD4+, detected by flow cytometry) were enriched from density gradient-enriched fraction of PBMC of the buffy coats using naïve CD4^+^T cell isolation kit II (Miltenyi Biotec). Isolated CD4^+^T cells were plated at 10^5^ cells/well and cocultured with 10^4^ prestimulated allogeneic MoLCs. After 5 days of coculture supernatants were harvested, cells were analyzed by flow cytometry.

### 4.5. Flow Cytometry

Before analysis cells were incubated with 20 µL anti-CD83 PE (HB15e), CD86 PE (2331), CCR7 PE (3D12), CD40 FITC (5C3), CD80 FITC (L307.4, all BD Biosciences, Heidelberg, Germany), HLA-DR FITC, PE (B8.12.2, Coulter, Krefeld, Germany) for 30 min on ice. After washing, cells were analyzed by FACSCalibur or LSRII (BD Biosciences) flow cytometers using CellQuestPro or FACSDiva software. Antibodies against TLR2 (TL2.1), TLR4 (HTA125), and TLR9 (eB72-1665, eBioscience, eBioscience, San Diego, USA) were used PE-coupled. For intracellular staining of TLR9, a fixation and permeabilization procedure was performed (Cytofix/Cytoperm, BD Biosciences). Cells in coculture were restimulated for 4 h of by 5 ng/mL PMA and 500 ng/mL ionomycin (Sigma-Aldrich, St. Louis, USA) in the presence of 1 µg/mL brefeldin A (BD Biosciences). After Cytofix/Cytoperm treatment (BD Biosciences) for 20 min cells were stained for intracellular expression by anti-IL-17A PE (eBio64DEC17), -IL-22 Alexa Fluor 647 (22URTI), -IL-9 PE (MH9D1, eBioscience), -IFN-γ APC (25723.11, BD Biosciences) for 30 min on ice. T cells isolated by magnetic activated cell sorting were routinely tested by 7-AAD, CD4 APC (RPA-T4), generated MoLCs by staining with HLA-DR APC (TÜ36, BD Biosciences) and Langerin PE (CD207, DCGM4, Coulter).

### 4.6. Cytokine Detection in MoLC and T Cell Supernatant

Culture supernatants were recovered before stimulation of intracellular cytokine production of the same samples. Release of IL-1β, IL-6, IL-12p70 (DuoSet, R&D Systems) and IL-23p19 (Biosource-Thermo Fisher Scientific, Waltham, USA) was determined by sandwich ELISA after 48 h in the supernatants of 1 × 10^6^ MoLC per ml. After 5 days protein levels of IFN-γ, IL-17A, IL-4, IL-9 and IL-22 (DuoSet, R&D Systems) were measured from supernatants of MoLCs:CD4^+^T cells cocultures. 

### 4.7. Statistical Analysis

Student’s two-tailed unpaired *t* test was used to calculate statistical significance. *p*-Values below 0.05 were considered significant.

## 5. Conclusions

Co-administration of bacterial LPS and PGN synergized in elevation of surface markers CD80, HLA-DR, CD40 and CCR7 on human MoLCs. In a coculture model with naïve CD4^+^T cells, MoLCs co-activated by a combination of PGN, LPS and DNA induced high level of IL-9, IL-22 and IFN-γ release and an increase of intracellular IL-17 in CD4^+^T cells.

## Figures and Tables

**Figure 1 ijms-20-01367-f001:**
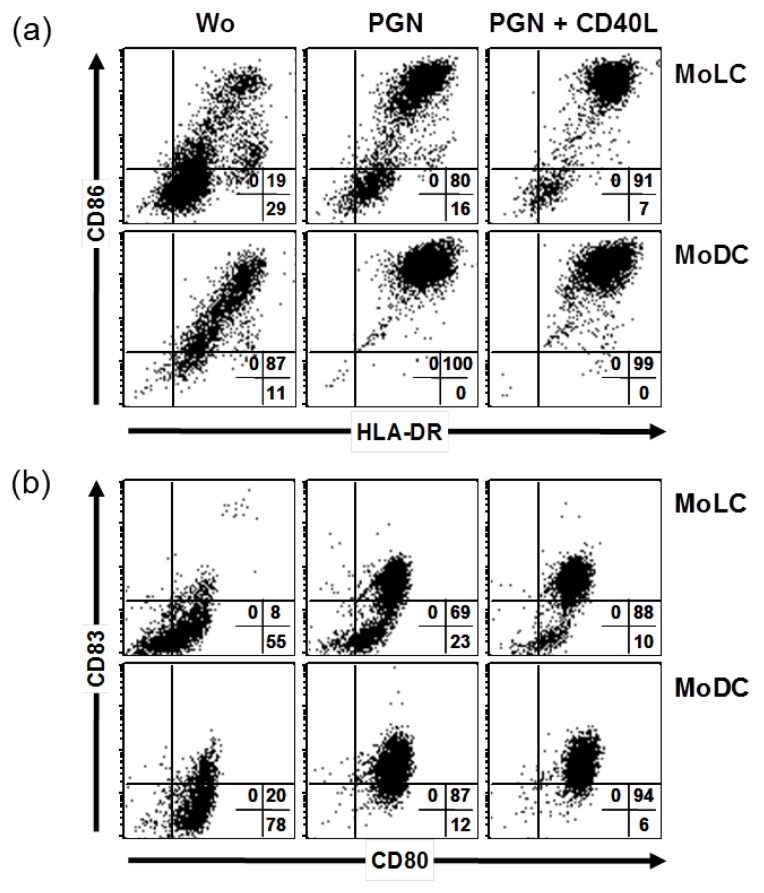
PGN and CD40 ligand synergize in upregulating maturation molecules on MoLCs. Derived from the same PBMC, MoDCs and MoLCs were stimulated for 48 h by TLR2 agonist 20 µg/mL PGN and 1 µg/mL trimer of CD40 ligand or remained without stimulus (Wo). (**a**) Expression of CD86 and HLA-DR and (**b**) expression of CD80 and CD83 was detected by flow cytometry on the surface of MoLCs (upper panel) and MoDCs (lower panel). Quadrants indicate staining with isotype controls, numbers percent positive cells. Dot plots shown are representative for four independent experiments with different donors.

**Figure 2 ijms-20-01367-f002:**
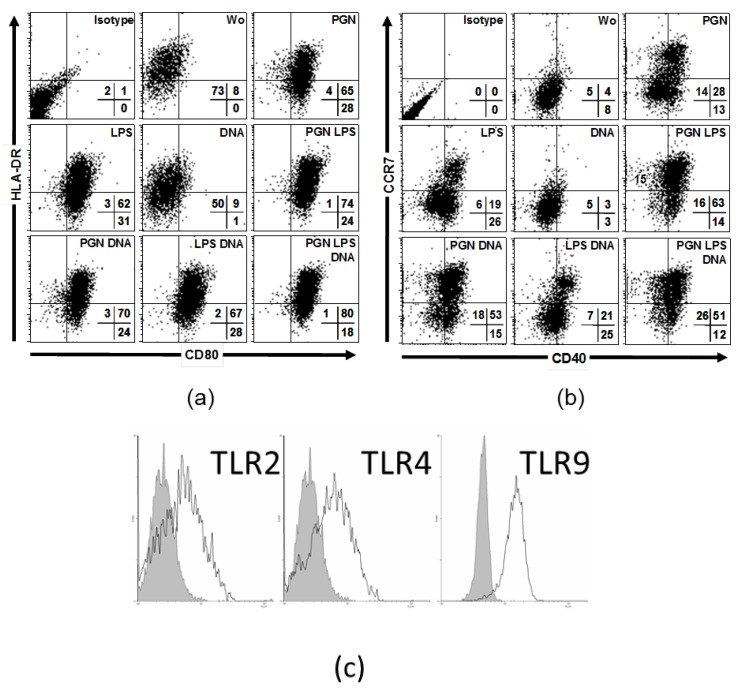
Triple TLR agonist activation increases LC-surface markers. MoLCs were activated by combinatory TLR2, TLR4 and TLR9 agonists: 20 µg/mL PGN, 1 µg/mL LPS, and 5 µg/mL DNA, respectively. (**a**) Dot plots show coexpression of CD80 and HLA-DR after 48 h. (**b**) Dot plots show coexpression of CD40 and CCR7 after 48 h. Quadrants indicate isotype controls, numbers percent positive cells. (**c**) Surface expression of TLR2, TLR4 and TLR9 in MoLCs. FACS staining of cells from one donor represents experiments from five donors with similar results.

**Figure 3 ijms-20-01367-f003:**
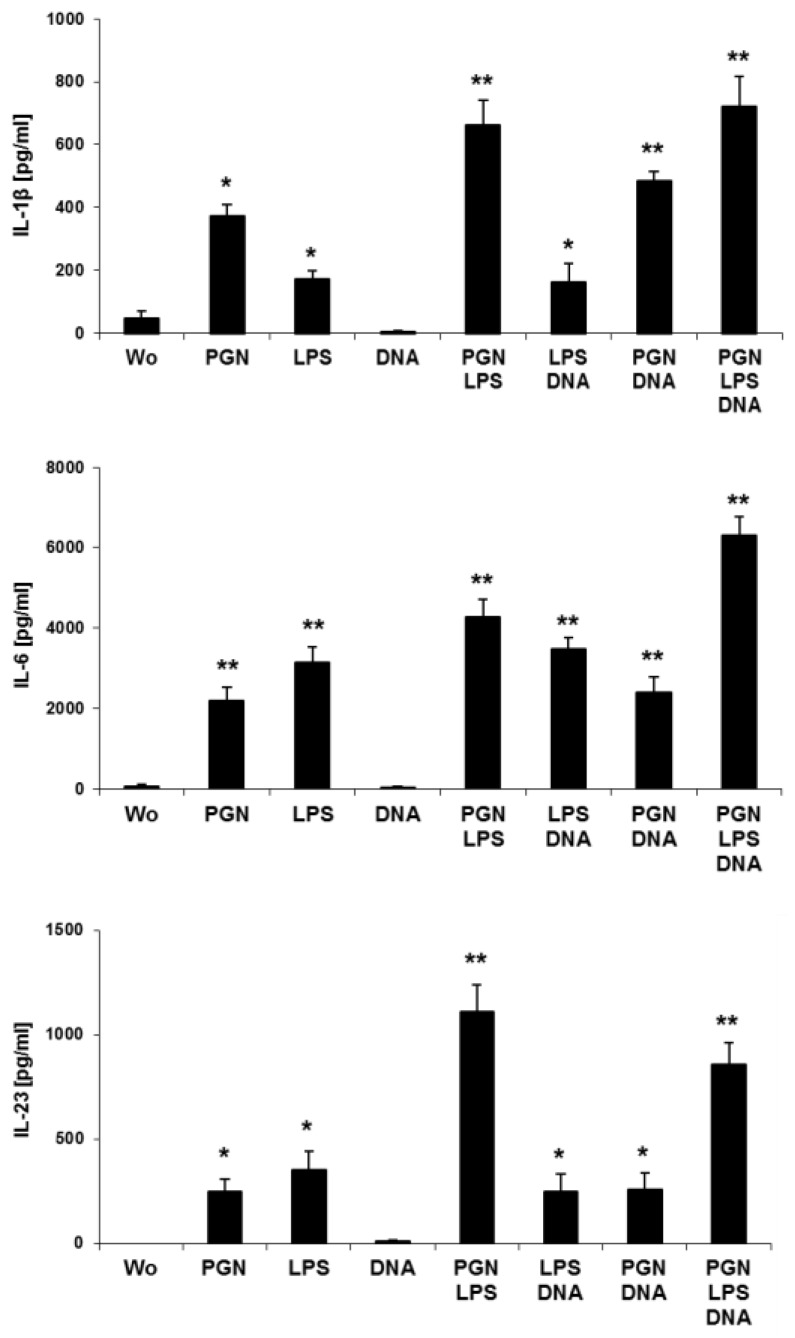
Triple TLR agonist activation increases crucial LC cytokines. MoLCs were activated by combinatory TLR2, TLR4 and TLR9 agonists: 20 µg/mL PGN, 1 µg/mL LPS, and 5 µg/mL DNA, respectively. The amounts of IL-1β, IL-6, and IL-23 were detected in supernatant after 48 h. ELISA bars show mean of cytokine values from four donors. Differences of values for cytokines after stimulation were analyzed for statistical significance and compared to control (without, wo); * *p* ≤ 0.05, ** *p* ≤ 0.01.

**Figure 4 ijms-20-01367-f004:**
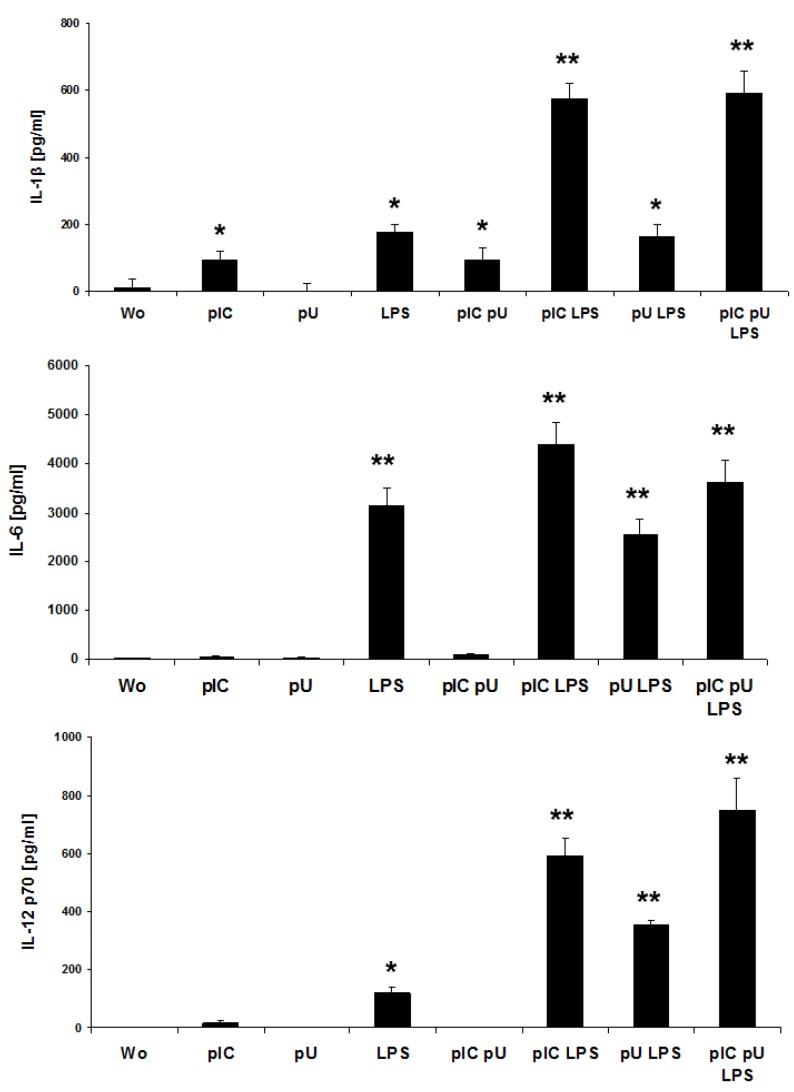
Analogs of nucleic acids elevate LPS-induced IL-1β and IL-12p70. MoLC were challenged with combinations of polyI:C (pIC, 5 µg/mL) and polyU (pU, 5 µg/mL) with LPS (1 µg/mL). Release of IL-1β, IL-6 and IL-12p70 was measured by ELISA after 48 h. Bars represent inter-experimental means ±SD of protein release in the media of MoLC from 4 different donors. Differences of values for cytokines after stimulation were analyzed for statistical significance and compared to control (without, wo); * *p* ≤ 0.05, ** *p* ≤ 0.01.

**Figure 5 ijms-20-01367-f005:**
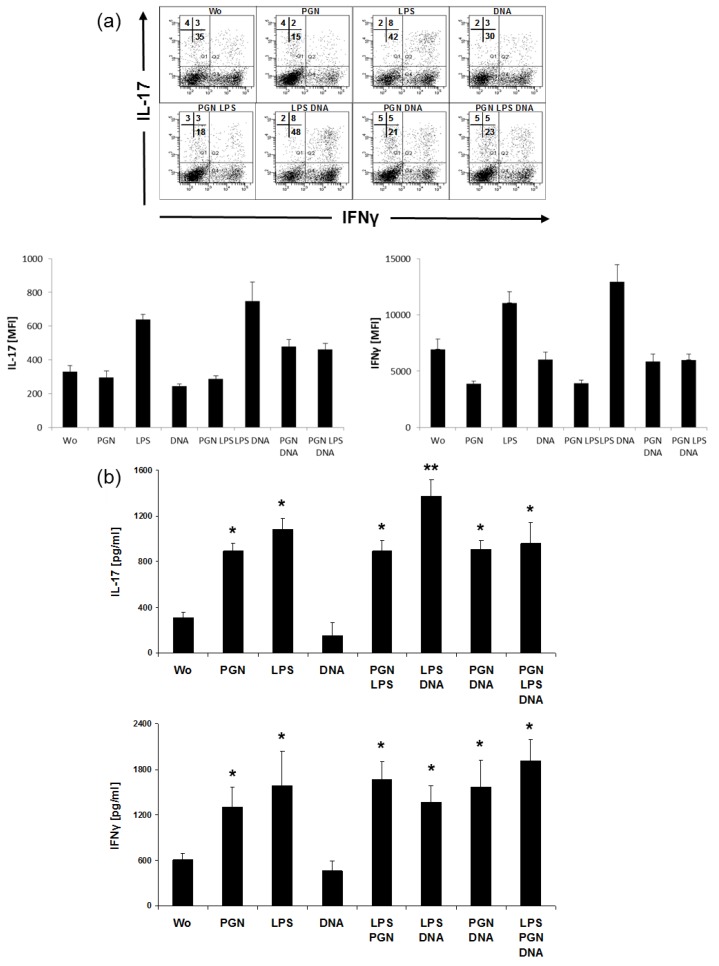
Intracellular and supernatant-derived IL-17 and IFN-γ in CD4^+^T cells from TLR-stimulated cocultures. MoLC were stimulated with single and combinations of TLR agonists 20 µg/mL PGN, 1 µg/mL LPS and 5 µg/mL DNA for 48 h. The stimuli were removed by washing and LCs further cocultured with allogeneic CD4^+^T cells. (**a**) Intracellular costaining of IL-17 and IFN-γ by flow cytometry in 5 days-cocultured cells. One experiment out of four with different donors and similar results is shown. Numbers represent the percentage positive cytokine expression among CD4^+^T cells; Bars indicate mean fluorescence intensity (MFI) ± SD from intracellular FACS for IL-17 and IFN-γ. (**b**) ELISA detects IL-17 and IFN-γ after 5 days in the supernatants. Bars represent results of experiments with four different donors (mean±SD) and include data from the same donor as used in flow cytometry (**a**). Differences of values for cytokines after stimulation were analyzed for statistical significance and compared to control (without, wo); * *p* ≤ 0.05, ** *p* ≤ 0.01.

**Figure 6 ijms-20-01367-f006:**
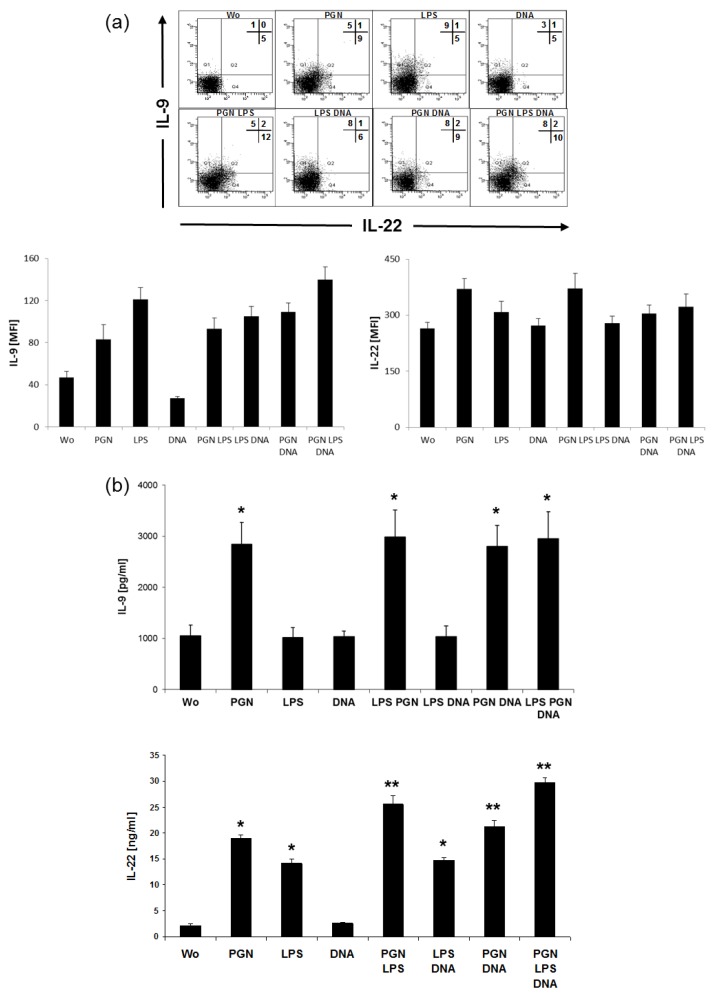
LPS, PGN and DNA synergize in inducing IL-22 and IL-9. After 48 h stimulation with 20 µg/mL PGN, 1 µg/mL LPS and 5 µg/mL DNA, stimuli were removed by washing and LCs further cocultured with allogeneic CD4^+^T cells. (**a**) Coexpression of IL-22 with IL-9 in cells cultured for 5 days. Shown is one experiment representing a total of tree with different donors, numbers indicate the percentage of positive cells, means of fluorescence intensity (mean FI) were analyzed from the dotplots. Bars indicate mean fluorescence intensity (MFI) ± SD from intracellular FACS for IL-9 and IL-22. (**b**) After additional 5 days, supernatants were analyzed for IL-9 and IL-22 by ELISA. Bars represent amounts of protein as inter-experimental means±SD of cytokine secretion in the supernatants of T cells from experiments using 5 independent MoLC and T cell donors. Differences of values for cytokines after stimulation were analyzed for statistical significance and compared to control (without, wo); * *p* ≤ 0.05, ** *p* ≤ 0.01.

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
