# Peer review of "In Vitro Induction of T Helper 17 Cells by Synergistic Activation of Human Monocyte-Derived Langerhans Cell-Like Cells with Bacterial Agonists"

_ijms, 2019, doi:10.3390/ijms20061367_

Round 1

Reviewer 1 Report

The authors have mostly satisfied my comments. However, they should consider putting the p values on the graph itself and for MFIs they should show a bar graph with error bars as well. I strongly believe that will improve the data presentation and suggest that the data behaves the same within the replicates. 

While the authors raise questions regarding different cytokines and their ability to polarize T cells, its widely accepted in immunology that certain cytokines-driven T cell polarization. I do understand the time constraints.

Author Response

Thank you for review and suggestions. As proposed, we added the p values on the graphs indicated by asterisks in Fig. 3-6. Four new graphs showing MFIs with error bars were added in Fig. 5 and Fig. 6.

Reviewer 2 Report

The authors answered the requested the changes and answered the different points raised by the reviewers. The modifications made the article more robust and improved the quality.

Author Response

Thank you for review and suggestions. We updated language and style.

This manuscript is a resubmission of an earlier submission. The following is a list of the peer review reports and author responses from that submission.

Round 1

Reviewer 1 Report

Authors should cite important papers in the field e.g. 

doi:10.1016/j.it.2013.07.006 and doi: 10.3389/fimmu.2018.01376

The authors do not explain the significance of IL-9 and IL-22 Th cells in the model. It seems Th17 are important during bacterial stimulation.

The authors must include quantification and MFI of activation markers for all FACs plots.

Please consider including P values for all the ELISA data.

Authors should consider performing a multiplex Luminex assay on the conditioned media to gain insight into the factors determining the T cell polarization

For the co-culture experiments were the CD4 T cells naive? Please also mention the source of these cells.

Reviewer 2 Report

The authors are demonstrating the ability of MoLCs to polarize the naïve CD4 T cells to TH17 cells. The authors use a protocol published in 2002 and the DC biology is much advanced with better characterization including additional markers, functional studies and transcriptomics. There is no compelling evidence to claim that the MoLCs are close to the bonafide LCs by phenotype or function or transcriptome profiling. Some of the studies from Caux et al, (PMID:9269763, PMID:8760823) clearly demonstrated the differential origin of LCS and MoDCs. As mentioned in the discussion, authors agree that most of the LCs generated from HSCs fails to induce the TH17 polarization.   The current observation may be due to the properties of MoDCs and nothing to do with LCs. I don’t think it’s right to compare them to LCs or claim the observations have any biological significance with LCs. The CD34+ derived LCs may be better model than MoDC derived LCs. It will be appropriate if the authors may dilute the claim from “human Langerhans cell-like cells” to “human Monocyte derived cells”, because the currently used model is not a thorough model to study for LC biology.

Major comments

TLR ligands synergize the DC maturation:

The authors assume the different ligands used for DC activations are specifically restricted to TLR ligands on DCs. The authors are not showing the level of expression of any TLR ligands on the in vitro generated DCs. It will be great to shows the expression of different TLRs on in vitro generated DCs. That may validate the presence of the TLRs against the ligands used and confirm the observed activation is TLR restricted and not mediated by other danger sensing pathways.  Why the authors are completely overlooking the possibility of peptidoglycans binding to NOD-like receptors or the plasmids as source of DNA is not activating the any NLRPS or AIM-2 etc or any other cytosolic DNA sensors.  The activation of NLRPS can also induce IL1- β, IL-6 and other inflammatory cytokines.  So authors has to show supporting data by FACS staining or qRT PCR  to validate their claims. The authors claims that the TLR expression demonstrated by different authors [16, 19, 25, 34, 35], but most of the in vitro generated LCs are from CD34+. It will be great to demonstrate in the Mo-LCs.

The role of TGF-β on the generation of IL-17 producing T cells: The authors demonstrate the production of different cytokines like IL-1 β, IL-6 and IL23, which are critical for the generation of Th17 cells. Some of the previous studies clearly demonstrated the critical role and requirement of TGF-β for the generation of Th17 cells (PMID: 18454150, PMID: 24865484). Authors should check the status of TGF- β production by Mo-LCs and include the data. What is the hypothesis the authors have the generation of TH-17 cells in the absence of TGF- β in human T cells.

Combination of LPS and IL-12p70: Why the DCs are fails to sense TLR3 ligands (Poly I: C / A: U) at the first phase? Does the TLR4 activation by LPS induce the TLR3 in DCs and that may be reason the combination have a stronger effect. Again whether the response is a TLR restricted sensing or other cytosolic sensors? It will be great to demonstrate the TLR3 expression with FACS staining or PCR at these conditions to strengthen the observations.

Fig6 a-b . The author’s claims that the LPS induced more IL9 producing T cells and the FACS data clearly demonstrate that. Whereas the figure 6b clearly shows that the level of IL-9 under LPS activations is equivalent to the unstimulated condition.  Whereas PGN activated MoLC: T cell condition produces 3X IL-9 in the culture.  The combination of LPS does not change the level of IL-9 in the culture. The data does not support the claim of authors and FACS data contradict the ELISA data. Authors should explain the difference or validate with additional experiments.  Similarly the level of IL-22 production by LPS is almost similar to PGN. So I don’t understand why the authors make a very ligand specific claim on T cell functional specialization without a solid and convincing supporting data.

Minor comments

Spelling: Line 89 : Stimulations, 115 devote?-  devoid , 198  activated LCs –activated MoLCs

Line 164 – Supplementary materials : I  Couldn’t see any supplementary materials attached with the manuscript